# Numerical study of aeolian vibration characteristics and fatigue life estimation of transmission conductors

**Jiaqiong Liu[1], Bo Yan[1]\*, Zheyue Mou[1], Yingbo Gao[2], Getu Niu[3], Xiaolin Li[3]**

**1** College of Aerospace Engineering, Chongqing University, Chongqing, China, **2** College of Mechanical and Vehicle Engineering, Chongqing University, Chongqing, China, **3** Inner Mongolia Power Economy and Technology Research Institute, Hohhot, China

\* boyan@cqu.edu.cn

**Data Availability Statement:** All relevant data are within the paper and its Supporting Information files.

## Abstract

The 2D computational fluid dynamics (CFD) model of transmission conductor is set up to simulate the aerodynamic forces varying with time on the conductor. Taking into account the geometrical nonlinearity of conductor lines, the finite element (FE) models of single span and two-span transmission lines discretized with beam elements are established. By means of the FE models, the aeolian vibrations of the conductor lines excited by the aerodynamic forces under different wind velocities are numerically simulated. The nonlinear resonant characteristics, the amplitude-frequency relations of the conductor lines during aeolian vibration are investigated, and the influences of the span length as well as the initial tension in conductors on the aeolian vibration characteristics are analyzed. Furthermore, a 3D FE model of a conductor segment and the suspension clamp is created to study the stress distributions of the 3D model corresponding to different lines during aeolian vibrations. Finally, based on the stress analysis of the 3D model, the fatigue lives of the transmission conductors during aeolian vibration under different wind velocities are estimated. The jump phenomenon induced by the nonlinear vibration is reflected by the numerical simulation considering the geometric nonlinearity, and it is found that the energy balance principle (EBP) overestimates the vibration amplitudes because it cannot take the influences of the geometrical nonlinearity and span length into account. The obtained results may provide some instructions for the prevention design of aeolian vibration.

## 1. Introduction

Overhead transmission lines are sensitive to the wind induced vibrations [1]. Aeolian vibration induced by the vortices shedding of air flow from the leeward of a conductor frequently takes place and usually leads to fatigue failure of the conductor strands. Fatigue failure of a conductor is frequently observed at the vicinity of the suspension clamp exit due to the complicated load and deformation of the conductor at this location. To control the aeolian vibration and extend operation lives of transmission lines, it is necessary to study the aeolian vibration characteristics and the fatigue lives of transmission conductors under aeolian vibration.

**Funding:** The research was funded by the Research Project of Inner Mongolia Power (Group) Co., LTD (No. 2020-39). The funders had no role in study design, data collection and analysis, decision to publish, or preparation of the manuscript.

**Competing interests:** The authors have declared that no competing interests exist.

There are mainly two kinds of theoretical calculation methods on aeolian vibrations: the energy balance principle (EBP) and the dynamic analysis method. Roughan [2] calculated the aeolian vibration amplitudes of conductors by means of the EBP and analyzed the effects of the vibration dampers on the vibration levels. Langlois and Legeron [3] developed a complete conductor-damper system and predicted the aeolian vibration amplitudes of the damped span with the EBP. Claren and Diana [4] studied the responses of the transmission lines subjected to the wind excitation forces by means of the mode superposition method without considering the bending stiffness of the conductor. Vecchiarelli [5] predicted the vertical amplitude of a single conductor span with a Stockbridge type damper during aeolian vibration using the iterative finite difference scheme, and indicated that the steady state of aeolian vibration is essentially in the form of a standing wave in the case without any damper. Kong et al. [6] developed a dynamic equation to analyze the aeolian vibration of a two-bundled conductor line with or without spacer dampers. However, the geometrical nonlinearity of the transmission lines during aeolian vibration is not considered in the aforementioned works.

In addition, there are some experimental works on the aeolian vibration of transmission lines. Diana and Falco [7] tested the forces transmitted by the wind to a cylinder and obtained the values of the energy introduced by the wind. Levesque et al. [8, 9] carried out an aeolian vibration test to measure the strain at key location on the conductor and analyzed the alternating stress amplitude of the conductors. But a wire of the conductor was removed to install the gauges, which may influence the stress distribution of the conductor. Kalombo et al. [10] studied the effects of the catenary parameter on the fatigue lives of overhead conductors with experiment data. Kubelwa et al. [11] evaluated the bending stress of envelope of both clamp edges at the spacer-damper and at the termination clamp with an indoor aeolian vibration test. Kubelwa et al. [12] also analyzed the relationship between the vibration amplitude and the bending stress of conductor lines under aeolian vibration using an experiment approach and statistical techniques. Reinke et al. [13] and Miranda et al. [14] evaluated the fatigue lives of two types of overhead transmission conductors and assessed the metallic and elastomeric clamps on the fatigue lives of these conductors. Godard et al. [15] evaluated the fatigue lives of overhead line conductors by means of the field experiment measurement data of conductors under aeolian vibrations. However, the experimental methods are too expensive to investigate the aeolian vibration characteristics of transmission lines with different structure parameters under different excitations.

In most of the FE models [16–18] used to investigate dynamic responses of transmission lines, the conductor lines are usually discretized by beam or cable elements to mitigate the model scale. With beam or cable model, the stress distribution in a real conductor cannot be reflected accurately due to the neglect of screwed strand structure of the conductor. In order to accurately predict the stress distribution in a conductor strand, 3D FE models [19, 20] of conductors with screwed structure near clamps were used in some numerical analyses. Lalonde et al. [21] analyzed the dynamic bending stresses of conductors at the clamp end and estimated the fatigue lives of the conductors subjected to wind loads by means of the FE model in which the steel and aluminum wires are simulated with beam elements. Lalonde et al. [22] proposed a FE model including both the conductor and suspension clamp to analyze the stress distributions of the conductor under wind induced vibrations. In this model, the steel and aluminum wires are simulated with beam elements too. Qi [23] used a 3D nonlinear FE model to analyze the stress of an ACSR conductor-clamp system under fretting fatigue conditions, and presented a multi-axial fatigue methodology to estimate the local fretting fatigue strength of the conductors, without taking into account the initial configuration of the conductor line.

In this paper, the aerodynamic forces on conductors are numerically determined by means of computational fluid dynamics (CFD) and the aeolian vibration characteristics of conductor

 

lines are investigated using the FE method. In addition, the stress distributions and fatigue lives of the transmission conductors under aeolian vibration are evaluated by a 3D refined FE model of the conductor segment and suspension clamp. The obtained results may provide foundation and instruction for the protection design of aeolian vibration and fatigue life estimation under aeolian vibration of transmission lines.

## 2. Aeolian vibration modelling of conductor lines

### 2.1. Aerodynamic forces on conductors

The aeolian vibration of a conductor is induced by the aerodynamic forces generated by the vortices which alternatively shed from the top and the bottom of the leeward side of the conductor [1]. Wind tunnel test is the direct method to determine the aerodynamic forces on the conductors under wind flow, but it is too much expensive. Fortunately, the CFD provides an available way to investigate the aerodynamic forces of the conductors in air flow.

**2.1.1 CFD model of air flow around a conductor.** The CFD has been employed to investigate the aerodynamic behavior of conductors in air flow by some authors, and 2D models are usually used in these works, ignoring the variation of aerodynamic forces along the line spans [17, 18, 24]. In general, the temperatures in transmission conductors in service are less than 70°C, so the air flow with different temperatures including 20, 30, 40, 50, 60 and 70°C around the conductor are analyzed. In all the cases the stable vortex streets and alternating aerodynamic forces are observed. The amplitudes of the lift coefficient corresponding to the six temperatures are 1.35, 1.32, 1.28, 1.25, 1.22 and 1.19 respectively. It is seen that the amplitude of the lift coefficient at 70°C is about 11% smaller than that at 20°C. It is very difficult to determine the temperature of the air around the conductor in real situation, but the temperature may be much lower than 70°. The aerodynamic forces in the case of 20°C are used in this paper.

The conductor JL/G1A-300/40 with diameter of 23.9 mm is studied in this paper. Aeolian vibration is generated by moderate wind in the range 0.5 to 7 m/s and it is most likely to occur when the wind direction is perpendicular to the conductor lines [1]. So, different incoming wind velocities in the range of 0.5 and 7 m/s are assumed and the wind attack angle is set to be 90°. The CFD model of air flow around a conductor is shown in Fig 1. Too large size of the

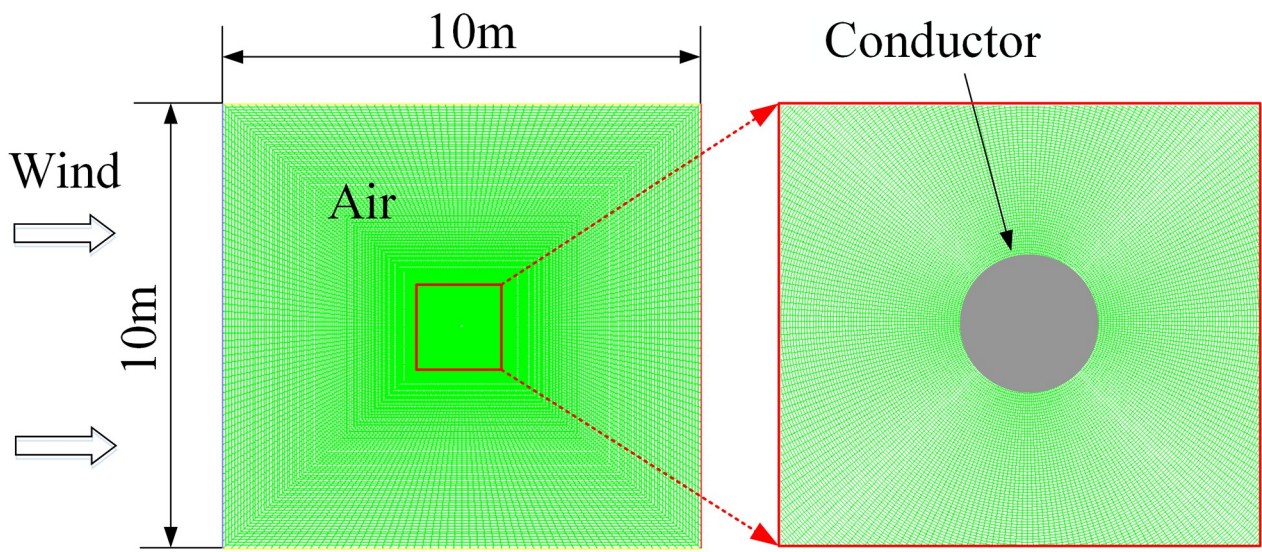

**Fig 1. CFD model of air flow around a conductor.**

domain will increase the computation cost and too small size will affect the air flow around the conductor. After several trials, it is found the size of the domain set to be 10 m × 10 m is suitable. The boundary conditions of the numerical model are set as follows: the left boundary is set to be the inlet of the air flow, the right is the outlet, and the vertical flow of the air on the top and bottom of the domain are constrained.

The FLUENT software is used to simulate the air flow around the conductor. The Spalart–Allmaras turbulent model is used in the analysis to depict the turbulence of air flow, and the Simple algorithm and Quick scheme with three-orders are chosen in the analysis. In addition, an increment of 0.001 s is set in the time integration for the transient analysis, and it is small enough to satisfy the accuracy requirement.

**2.1.2 Aerodynamic forces on conductor.** When wind velocity exceeds the limit of laminar flow, vortices will generate at the leeward of the conductor cross-section, inducing alternative lift and drag on the conductor surface. The aerodynamic forces on a conductor are defined as

$$F_L = \frac{1}{2}\rho v^2 C_L l d \quad F_D = \frac{1}{2}\rho v^2 C_D l d \tag{1}$$

where $F_L$ and $F_D$ are the lift and the drag respectively; $C_L$ and $C_D$ are the lift coefficient and the drag coefficient respectively; $\rho$ the density of the air; $v$ the wind velocity; $l$ the length of the model and $d$ the diameter of the conductor.

The velocity contours of air flow around the conductor in the case with wind velocity of 2 m/s at two typical times are shown in Fig 2, in which the alternating Karmen vortex street behind the conductor is observed.

The time history curves of the aerodynamic coefficients in the case with wind velocity of 2 m/s are shown in Fig 3. It is seen that the coefficients oscillate with time at the initial stage and tend to be steady after a short period. The mean value of the lift coefficient is near 0, while the mean value of the drag coefficient is 1.35, which is consistent with that presented in [25].

The frequency and amplitude of the lift changing with wind velocity of the conductor are shown in Fig 4, from which it is seen that the frequency increases approximately linearly with the wind velocity, but the amplitude increases nonlinearly. From Eq (1) it is known that the lift is proportional to the square of wind velocity.

The Strouhal number $S$ of the lift is determined by

$$S = f_s d / v \tag{2}$$

where $f_s$ is the vibration frequency caused by the vortex shedding. The calculated Strouhal numbers at different wind velocities range from 0.21 to 0.23, which is consistent with the work

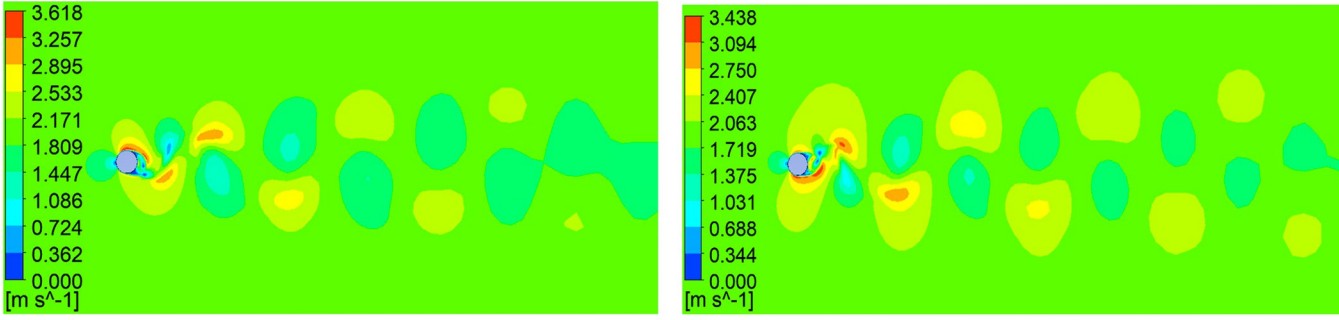

**Fig 2. Velocity contour of air flow around the conductor with wind velocity of 2 m/s at two typical times.** (A) 2.94 s; (B) 3 s.

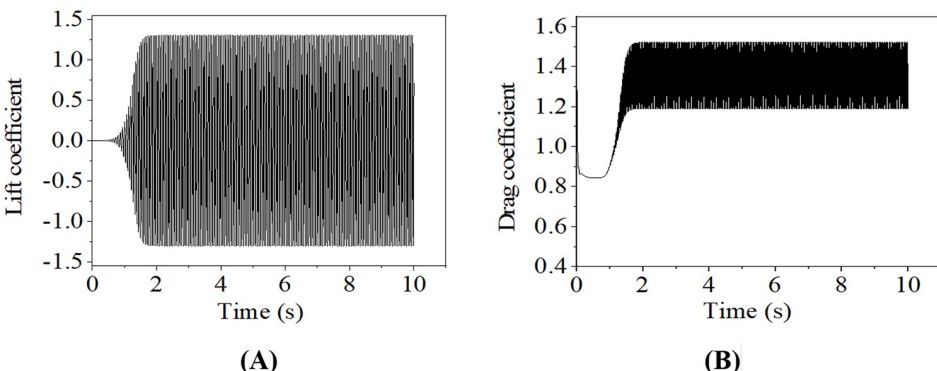

**Fig 3. Aerodynamic coefficients of conductor with wind velocity of 2 m/s.** (A) Lift coefficient; (B) Drag coefficient.

of Ref. [26]. The aerodynamic forces determined by the CFD model can be used to simulate the aeolian vibration of the transmission lines.

## 2.2. FE modeling method of aeolian vibration of conductor lines

It is assumed that the wind is uniformly applied on the conductor line with the same velocity and perpendicular to the line; the geometrical nonlinearity of large displacement and small strain is taken into account; and the influence of the towers on the aeolian vibration of the conductors is ignored.

The ABAQUS software is used to simulate the aeolian vibration of transmission conductors, and a typical FE model of the conductor line is shown as in Fig 5. The conductor is discretized by spatial beam elements and the geometrical nonlinearity of large displacement and small strain is taken into account. The two ends of the conductor line are fixed and the alternative wind excitation forces $F_L$ and $F_D$ obtained from the CFD simulations are applied on the conductor line along the line. Mesh size is determined based on the mesh check to arrive at convergence, and it is found that the mesh size of 0.1 m is enough to obtain convergent results. Finally, the damping ratio of the conductor is set to be 0.05% of critical damping [27].

For a cylinder with specific diameter, the vortex shedding rate depends on the wind velocity and aeolian vibration may take place under the excitation of the aerodynamic forces changing

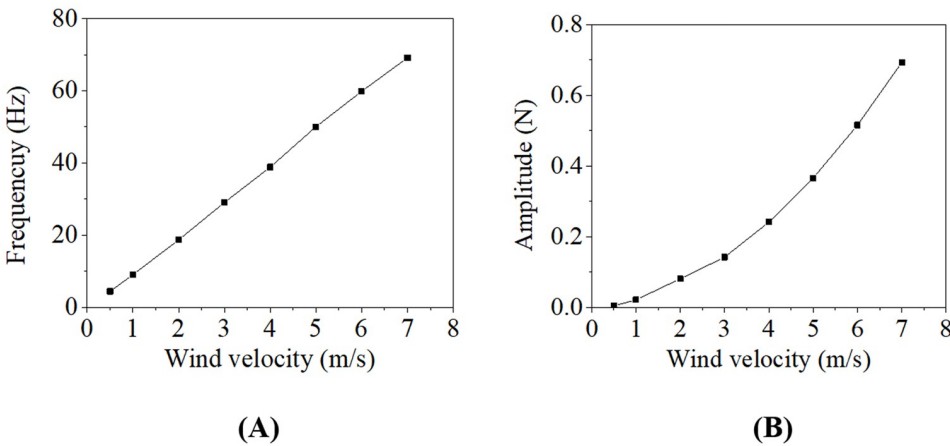

**Fig 4. Frequency and amplitude of lift versus wind velocity.** (A) Frequency; (B) Amplitude.

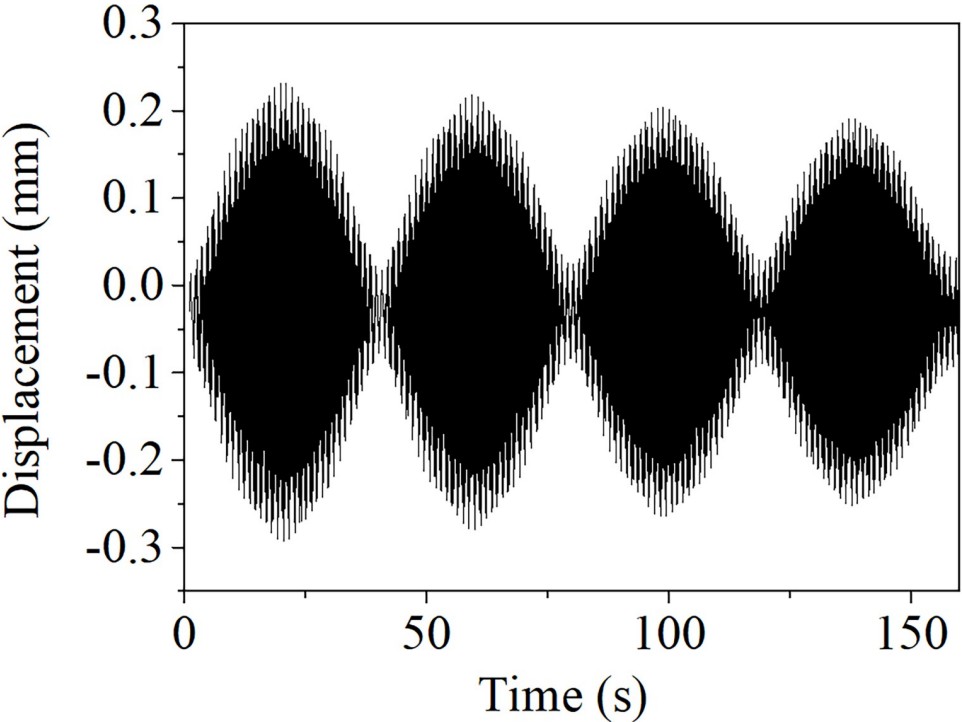

**Fig 5. FE model of single span transmission conductor line.**

with specific frequency. Due to the geometric nonlinearity of large displacement and small strain of transmission lines, when resonant vibration takes place, the aeolian vibration frequency of the transmission line may be not exactly equal to the linear natural frequency of the line. To excite an aeolian vibration and find out the resonant frequency, a process called as "frequency sweep" is performed, in which the wind excitation frequency is continuously changed around a natural frequency of the line with an increment of 0.0001 Hz during simulation. For the conductor lines, aeolian vibrations with frequencies closed to several linear natural frequencies are numerically simulated, and the results will be discussed in next section.

In the process of "frequency sweep", there are beat waves on the transmission lines when the excitation frequencies are close to the resonant frequency of the transmission lines, which is a typical characteristic of the aeolian vibration [1]. A typical beat wave obtained from the FE model is shown in Fig 6.

**Fig 6. A beat wave in aeolian vibration.**

It is necessary to verify the CFD and FE model by comparing the energy that the wind input to the transmission lines with the previous experiments. The wind energy input per unit length $E$ is calculated by the Eq (3) [5]:

$$E = \pi F f_c y_0 \tag{3}$$

where $F$ is the amplitude of the force per unit length of the conductor, $y_0$ the vibration amplitude of the conductor, and $f_c$ the resonant frequency of the conductor. The relationship between the relative amplitude and the wind input power is plotted in Fig 7, where the relative amplitude is defined as $y_0/d$ and the wind power input is defined as $E/(f_c^3 d^4)$. The solid blue line is the wind input power of the 100 m-span line with initial tension stress of 84.38 MPa obtained from the FE model and the dotted lines from the experiments [7, 28]. The data obtained from FE model approximately locate between two experimental ones, which verifies the validation of the numerical models.

The numerical simulations were carried out on a workstation of Dell T7920 with CPU of Intel (R) Xeon(R) Gold 6140M 2.30GHz and RAM of 256G. To excite the aeolian vibration of a line at a frequency closed to a lower order of natural frequency, the process of sweeping frequency and arriving at steady oscillation spent about 30 hours, while at a frequency closed to a higher order of natural frequency, it spent about 2 weeks in some cases. To speed up the simulation efficiency, a batch processing file was written using Python and many jobs were running at the same time.

## 3. Aeolian vibration characteristics of conductor lines

### 3.1. Parameters of conductor lines

The conductor JL/G1A-300/40 with a diameter of 23.9 mm is chosen to analyze the aeolian vibration. The Young's modulus, Poisson's ratio and density of the conductor are 73000 MPa,

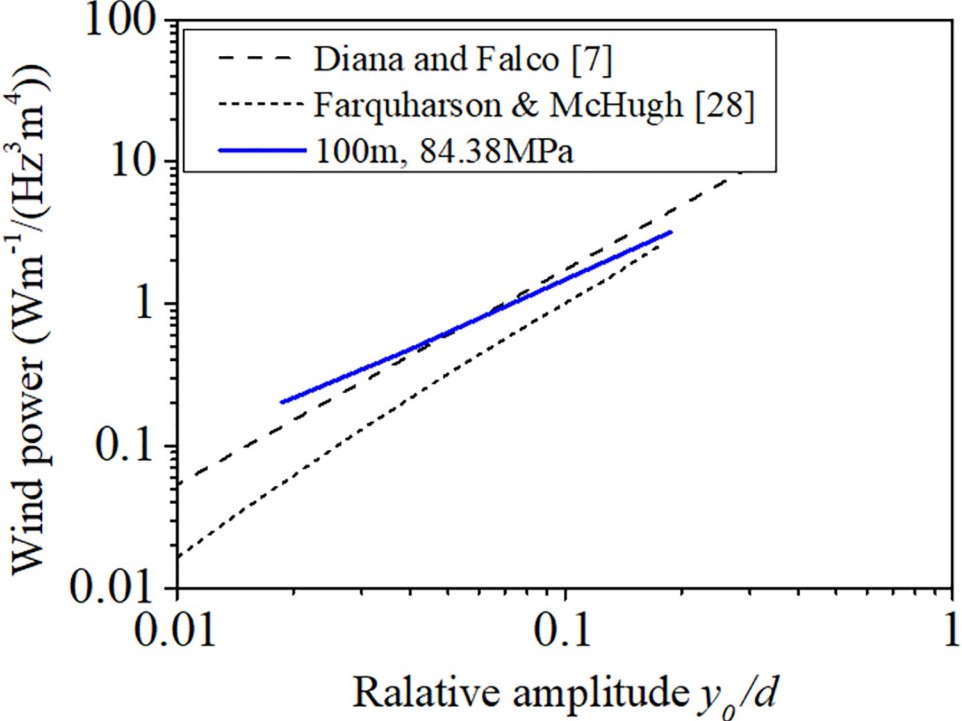

**Fig 7. Wind input powers of FE model and experiments [7, 28].**

Table 1. **Parameters of single span transmission lines.**

| Case number | 100–1 | 100–2 | 150–1 | 150–2 | 150–3 | 200 | 250 | 300 |
|---|---|---|---|---|---|---|---|---|
| Span length (m) | 100 | 100 | 150 | 150 | 150 | 200 | 250 | 300 |
| Initial stress (MPa) | 84.38 | 90.82 | 61.11 | 80.69 | 90.76 | 86.15 | 81.88 | 72.68 |

0.3 and 3342 kg/m$^3$ respectively. Single span lines are discussed in this section. The aeolian vibration of a transmission line depends not only on the wind velocities, but also on the natural frequencies and modes of the line, which are related with the structural parameters such as span length, initial tension or sag of the line. The effects of these parameters on the aeolian vibration characteristics will be numerically investigated in this section.

Eight cases of single span lines with different span lengths and initial tension stresses in the conductors are numerically investigated and the different parameters are listed in Table 1.

### 3.2. Resonant characteristics of aeolian vibration

To analyze the resonant characteristics of aeolian vibration, the line 100–2, i.e. 100m-span line with initial tension stress of 90.82 MPa, is discussed in this section. The aeolian vibrations of the conductor line closed to four linear natural frequencies are simulated and the final resonant frequencies are respectively 7.4271 Hz, 15.6536 Hz, 32.6600 Hz and 46.7070 Hz, which are not the linear natural frequencies of the line, 7.4394 Hz, 15.708 Hz, 32.702 Hz and 46.761 Hz respectively, due to the geometrical nonlinearity. Using Eq (2), the wind velocities corresponding to the excited aeolian vibrations can be estimated, and the velocities are 0.81 m/s, 1.70 m/s, 3.55 m/s and 5.07 m/s respectively.

The vertical displacement time history at midpoint of the line 100–2 under each resonant excitation frequency $f_e$ is shown in Fig 8, and the frequency spectra of the displacement responses are shown in Fig 9. It is seen that the displacements oscillate with time at the initial stage and tend to be steady after a short period. In other words, the vibrations of the conductor line under all excitation frequencies are monofrequent after steady state arrives, which is consistent with the result obtained by Vecchiarelli [5], but the multi-frequency phenomenon in the initial stage of the aeolian vibration is not reflected in the work of Ref [5]. It is noted that a lower order of mode is excited in the initial time period, but gradually decreases and finally disappears in each curve. This lower order of mode is a free vibration excited by the initial application of the load and disappears after steady vibration state arrives. Due to the small damping of the conductor line, it spends about two weeks to arrive at the steady state for the numerical simulation of the aeolian vibration in Fig 8(C).

Resonance may take place as the wind excitation frequency is close to one natural frequency of the conductor. To capture the resonance frequency, sweeping frequency is carried out near a natural frequency of the conductor line closed to the excitation frequency corresponding a wind velocity. The variations of amplitudes around resonant frequencies of case number 100–2 in several sweeping ranges of frequencies are shown in Fig 10. It is seen that in the sweeping range of frequency, the vertical vibration amplitude of the conductor line gradually increases with frequency initially and suddenly jumps down after arrives at a peak value. This may be due to the 'jump phenomenon' for primary resonance of a nonlinear system.

The aeolian vibration modes of the single span line under different resonant frequencies are shown in Fig 11. It is seen that the largest amplitude occurs at the frequency of 7.4271 Hz, and after that the higher the frequency, the smaller the amplitude. It is noted that all the wind excitation frequencies are close to the linear natural frequencies of the line, 7.4394 Hz, 15.708

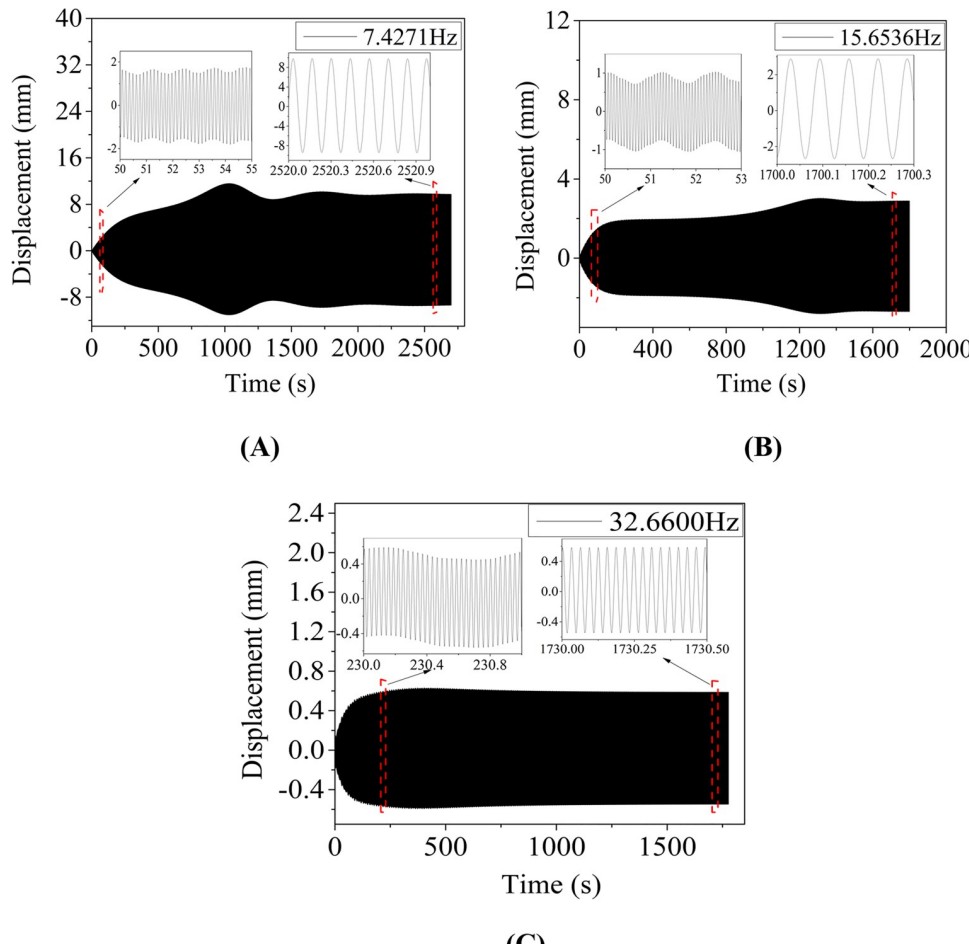

**Fig 8. Vertical displacement histories at midpoint of 100 m-span line with initial tension stress of 90.82 MPa under different resonant excitation frequencies.** (A) $f_e$ = 7.4271 Hz; (B) $f_e$ = 15.6536 Hz; (C) $f_e$ = 32.6600 Hz.

Hz, 32.702 Hz and 46.761 Hz respectively, and the aeolian vibration modes are the same with the corresponding natural modes.

## 3.3. Resonant amplitude-frequency curves

The resonant amplitude-frequency curves of the conductor lines listed in Table 1 are shown in Fig 12. The curves obtained by the numerical simulation and the EBP for the cases 100–2, 150–1 and 150–3 are compared in Fig 12(A). The wind energy input used in EBP is calculated by Eq (3) and the energy dissipated function of the conductor self-damping is from reference [2]. It is seen from Fig 12(A) that the amplitudes obtained by the EBP and the numerical method decrease with frequency due to the increase of the damping with the frequency. But the vibration amplitudes determined by the EBP are larger than those by the numerical simulation. On the other hand, the EBP does not take into account the effect of span length on the aeolian vibration. It can be seen from Fig 12(A) that when initial tension in the conductor is the same, the span length obviously affects the amplitude. The longer the span, the smaller the vibration amplitude of the conductor lines with the same initial tension under the same excitation frequency.

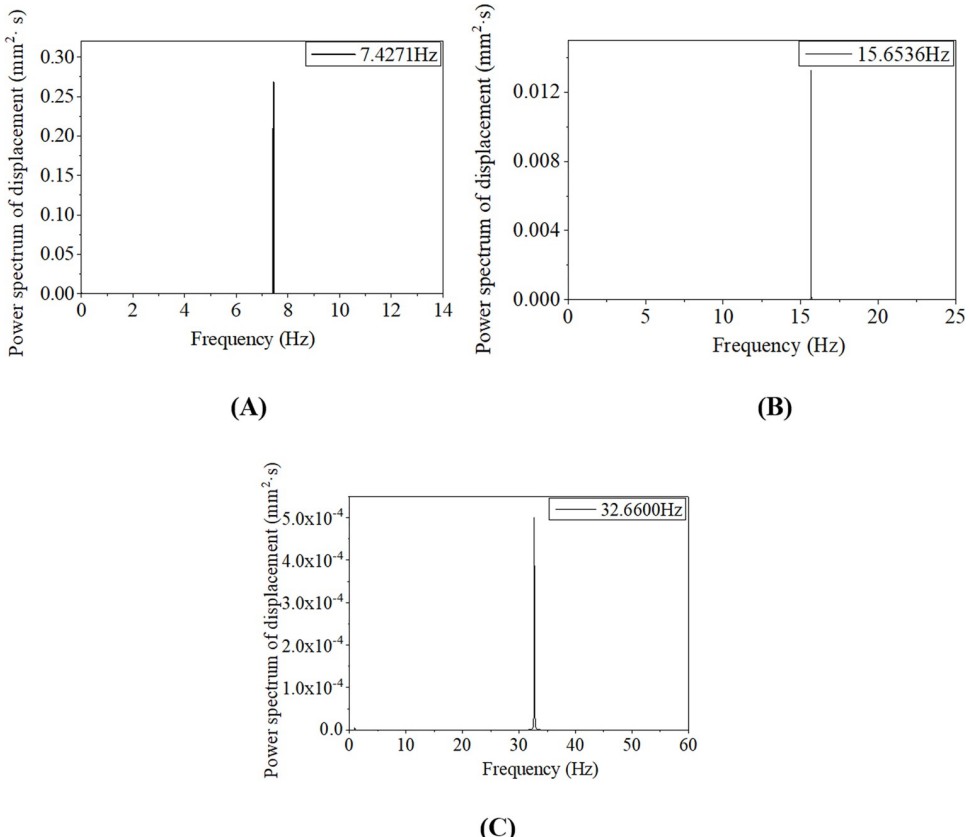

**Fig 9. Frequency spectra of vertical displacement histories at midpoint of 100 m-span line with initial tension stress of 90.82 MPa under different resonant excitation frequencies.** (A) $f_e$ = 7.4271 Hz; (B) $f_e$ = 15.6536 Hz; (C) $f_e$ = 32.6600 Hz.

The numerically determined resonant amplitude-frequency curves of the conductor lines listed in Table 1 are shown in Fig 12(B), from which it is known that the span length and initial tension stress in the conductors obviously affect the amplitude-frequency relation of aeolian vibration, which will be discussed in the next section.

## 3.4. Effects of initial tension and span length on aeolian vibration characteristics

As shown in Fig 12(B), the resonant amplitude-frequency curves of the 100 m-span lines with two different tensions of 84.38 MPa and 90.82 MPa are different, and the curves of the 150 m-span lines with three different tensions, 61.11 MPa, 80.69 MPa and 90.76 MPa, are also different. It is seen that the larger the initial tension of the conductor, the larger the vibration amplitude of the conductor under the same excitation frequency, which is consistent with those by the CIGRE [1].

On the other hand, the resonant amplitude-frequency curves of the 100 m-span and 150 m-span lines with nearly the same tension, respectively 90.82 MPa and 90.76 MPa, are obviously different, and those curves of the 100 m-span and 200 m-span lines with initial tensions, respectively 84.38 MPa and 86.15 MPa, are different too. It is apparent that the longer the span, the smaller the vibration amplitude of the conductor lines with the same initial tension under the same excitation frequency.

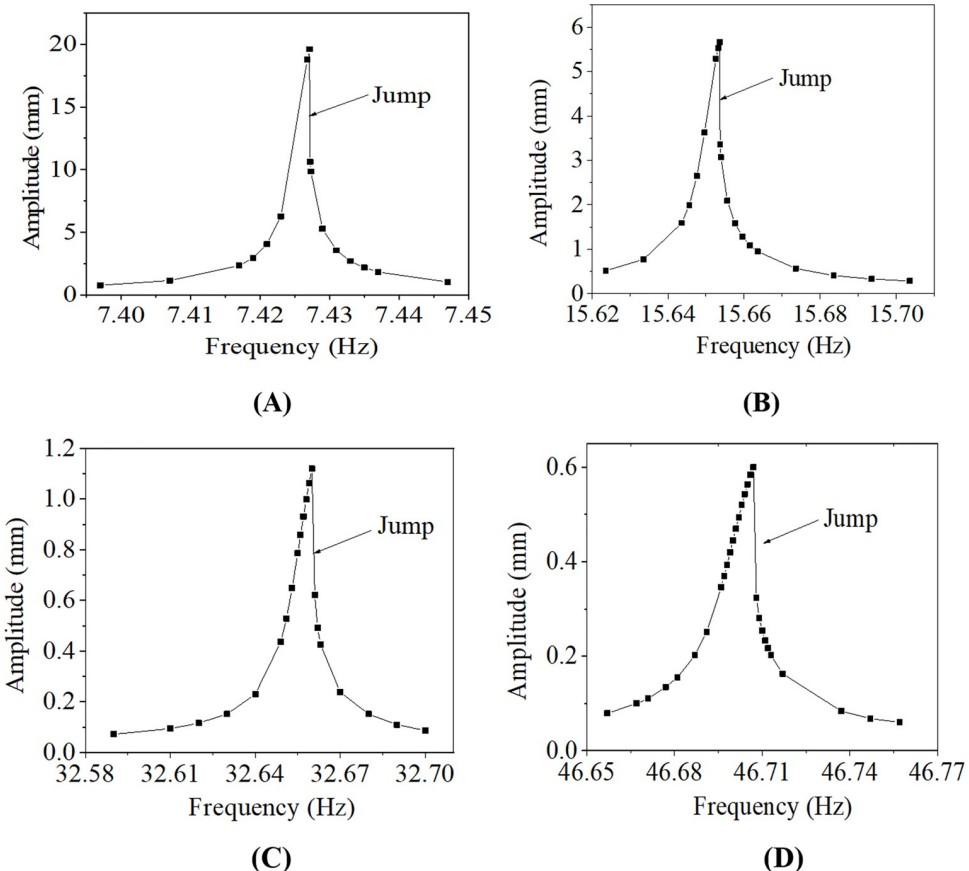

**Fig 10. Amplitude variations around resonance frequencies of single 100 m-span line with initial tension stress of 90.82 MPa.** (A) 7.3970 Hz~7.4470 Hz; (B) 15.6236 Hz~15.7036 Hz; (C) 32.5900 Hz~32.7000 Hz; (D) 46.6570 Hz~46.7570 Hz.

Therefore, for a line with specific span length, decreasing the initial tension in the conductor may mitigate the amplitude if the aeolian vibration takes place and in turn extends the fatigue life of the conductor. Moreover, the aeolian vibration amplitude of a conductor line with longer span is smaller than that of a line with short span.

## 4. 3D stress analysis and fatigue life estimation of conductors

It is well known that the conductor at the suspension clamp exit is subjected to a dynamic tension, to a bending force caused by the deflection of the conductor and to a clamping force exerted by the clamp. These alternatively changing forces may lead to fatigue failure at contact points between the wires of the stranded conductor. Conductor breakage induced by fatigue of aluminum wires is one of the most frequently observed phenomenon, and aeolian vibration is the main reason leading to fatigue failure of the wires in a conductor. Fatigue failure of aluminum wires in conductors due to aeolian vibration are shown as in Fig 13.

Fatigue test is a direct way to estimate the fatigue life of a conductor under aeolian vibration, but it is very time-consuming and expensive. Numerical simulation provides an efficient way to estimate the fatigue life. However, the stress distribution in a conductor cannot be reflected accurately when the conductor is simulated by simplified beam elements without considering the effects of the stranded details on the conductors. So, it is necessary to simulate the stress distribution of a conductor by using a 3D FE model.

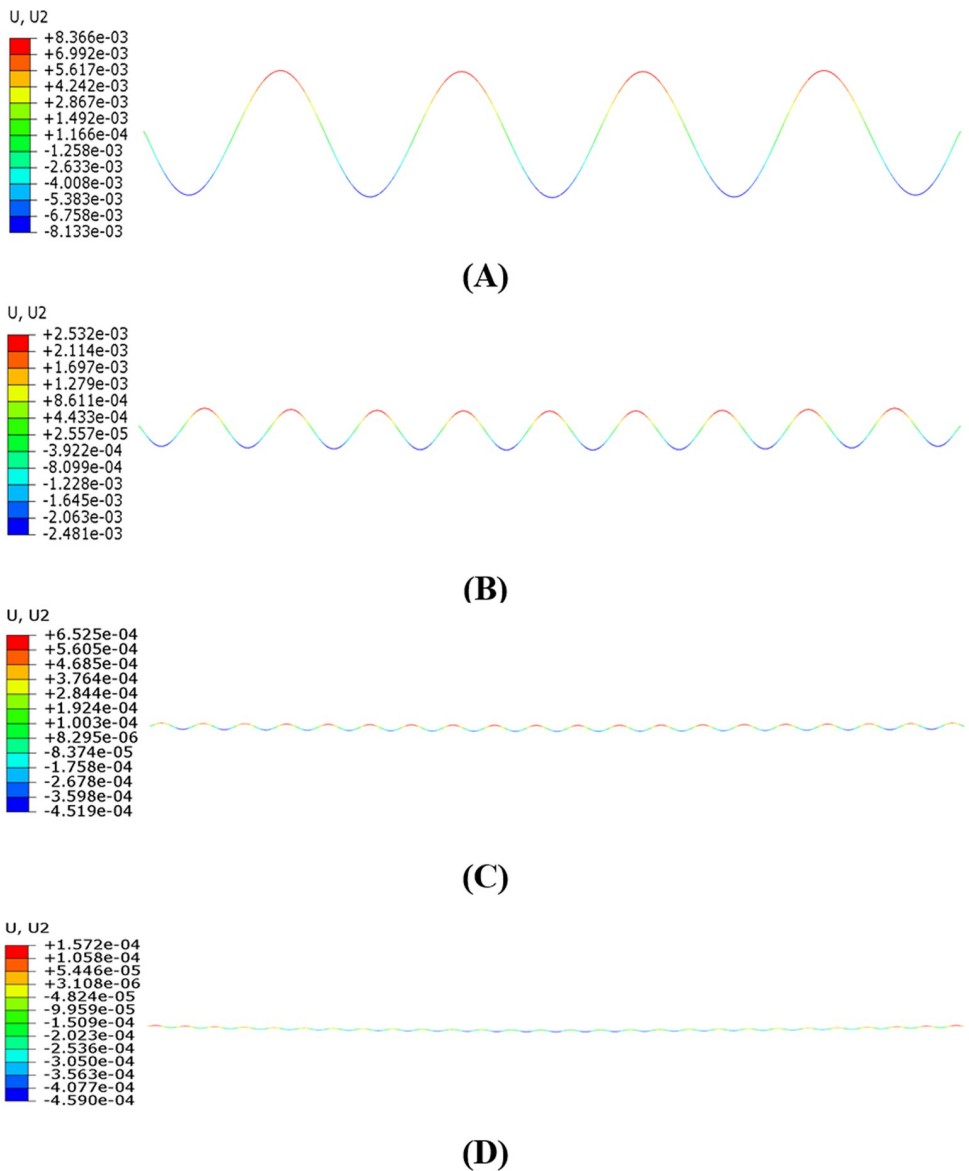

**Fig 11. Steady aeolian vibration modes of single span line with 100 m-span length and initial tension stress of 90.82 MPa at various excitation frequencies (Displacements magnified 800 times).** (A) $f_e$ = 7.4271 Hz; (B) $f_e$ = 15.6536 Hz; (C) $f_e$ = 32.6600 Hz; (D) $f_e$ = 46.7070 Hz.

## 4.1 Aeolian vibrations of two-span conductor lines

To study the aeolian vibration of a conductor at the suspension clamp exit, the aeolian vibration of a two-span transmission line should be simulated. Eight two-span transmission lines, in which the length of each span and the initial tension in the conductors are the same with those of the single-span lines as shown in Table 1, are investigated in this paper.

In this section, the aeolian vibration characteristics of the two-span line with a length of 100 m in each span and initial tension stress of 90.82 MPa around four natural frequencies of the line are analyzed, based on the dynamic responses of the line numerically simulated by means of the method presented in Section 2.2. Due to the geometrical nonlinearity of the line, the resonant frequencies determined by the "frequency sweep" are respectively 7.4161 Hz, 15.6314

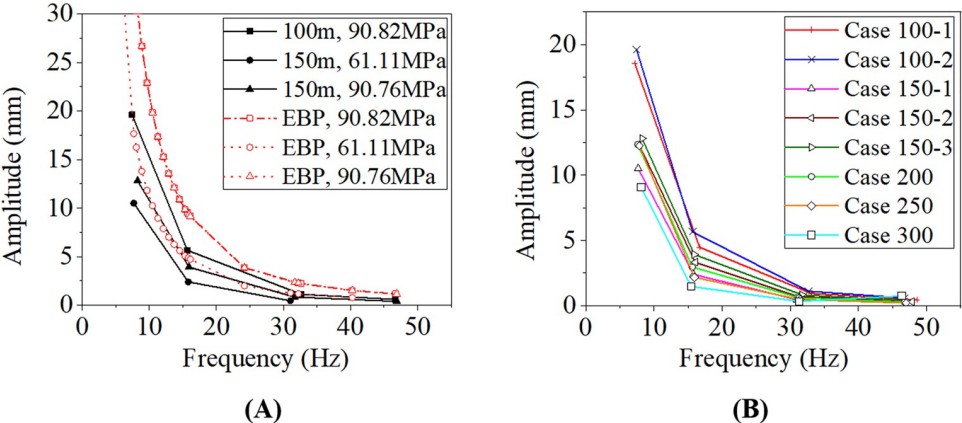

**Fig 12. Resonant amplitude-frequency curves of conductor lines under aeolian vibrations.** (A) Comparisons between numerical simulation and EBP; (B) Amplitude-frequency curves of different cases.

Hz, 31.4790 Hz and 48.5590 Hz. The wind velocities corresponding to the exciting frequencies estimated by using Eq (2), are respectively 0.81 m/s, 1.70 m/s, 3.42 m/s and 5.28 m/s.

The vertical displacement time history at midpoint of a span under wind excitation frequency of 7.4161 Hz is shown in Fig 14. It is seen that steady vibration arrives after 1800 s and the vertical vibration peak-to-peak amplitude is 18.01 mm.

The aeolian vibration modes of the two-span line under different wind excitation frequencies are shown in Fig 15. Similar to the single span line, the largest vibration amplitude occurs at the excitation frequency of 7.4161 Hz, and after that the higher the frequency, the smaller the amplitude. It is noted that all the wind excitation frequencies are close to the linear natural frequencies of the line, 7.4287 Hz, 15.686 Hz, 31.515 Hz and 48.611 Hz, and the aeolian vibration modes are the same with the corresponding natural modes.

### 4.2 3D FE modeling of conductor and suspension clamp

**4.2.1 Geometrical model of conductor and suspension clamp.** Based on the literature [29], the fatigue life of an energized conductor with higher temperature is slightly larger than that of a nonelectrical conductor with lower temperature. In this study, the effect of the temperature on the fatigue lives of the conductors is ignored. And the influences of heat transfer between the conductor and the surroundings on the vibration characteristics and fatigue lives of the conductors are also ignored. The stranded structure of conductor JL/G1A-300/40 is shown in Fig 16 and its structural parameters are listed in Table 2. The XGU-4F suspension

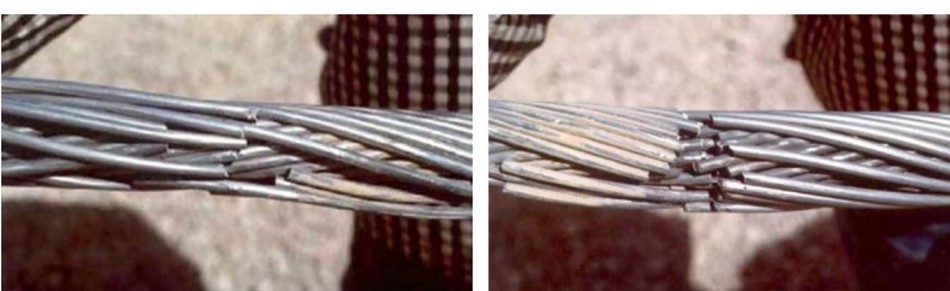

**Fig 13. Fatigue failure of aluminum wires due to aeolian vibration [1].**

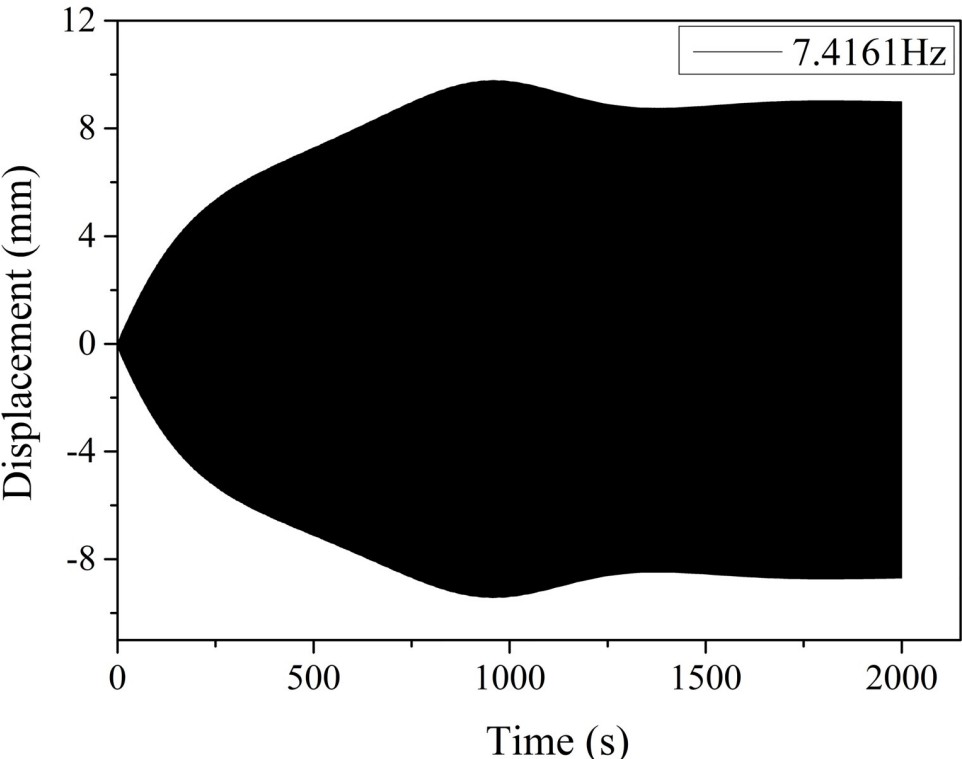

**Fig 14. Vertical displacement history at midpoint of one span of two-span line under excitation frequency of 7.4161 Hz.**

clamp is chosen based on the diameter of the conductor and the standard of the hardware fittings, and its structure and dimension are shown in Fig 17.

The displacement of the suspended insulator string is very small compared with that of the conductor during aeolian vibration according to the numerical results in Section 4.1. Therefore, it is assumed that the suspended insulator string is fixed and only a half of the suspension clamp is included in the analytical model, as shown in Fig 18, which is built with the Solid-Works software. The total length of the conductor segment in the model is 215 mm.

The sag of the conductor at the clamp exit can be determined by the initial configuration of the conductor line under self-weight. In the case without elevation difference, the sag $y_s$ of the conductor line at a point can be determined by [30]

$$y_s = -\frac{p_0 l}{2KT}\left[\text{sh}Kx - \frac{\text{ch}Kx - 1}{\text{th}\frac{Kl}{2}}\right] + \frac{p_0 x}{2T}(l_s - x) \qquad (4)$$

where $l_s$ is the span length, $p_0$ the load per unit length, which is generated by the self-weight here, $x$ is the horizontal coordinate of the point with the origin of the coordinate system at the left suspension point of the conductor, and $K = \sqrt{\frac{T}{EI}}$, in which $T$ is the tension of the conductor and $EI$ the bending stiffness of the conductor [31, 32]. The sags at the right end of the conductor segment, as shown in Fig 18, are listed in Table 3, which corresponds to the 8 transmission lines determined by Eq (4).

**4.2.2 FE model of conductor and suspension clamp.** The 3D geometrical models of the conductor segment and the suspension clamp generated in SolidWorks are imported into the ABAQUS/CAE to establish the FE model, as shown in Fig 19(A). The conductor segment is

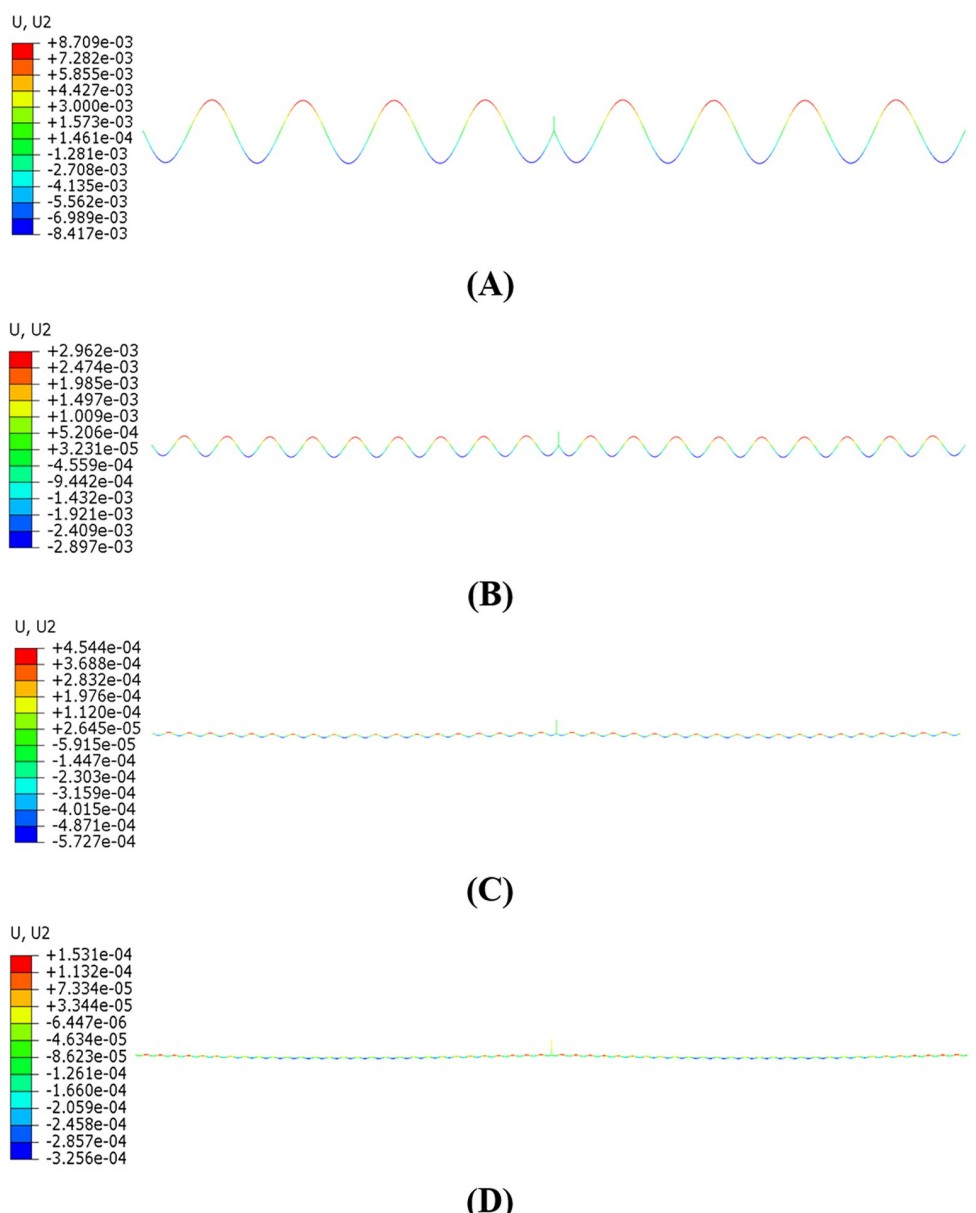

**Fig 15. Aeolian vibration modes of two-span line with 100 m-span length and initial tension stress of 90.82 MPa under various wind excitation frequencies (Displacements magnified 800 times).** (A) $f_e$ = 7.4161 Hz; (B) $f_e$ = 15.6314 Hz; (C) $f_e$ = 31.4790 Hz; (D) $f_e$ = 48.5590 Hz.

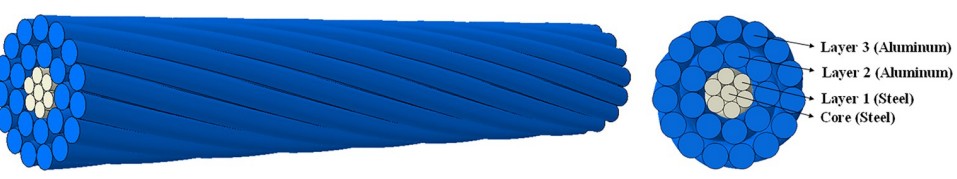

**Fig 16. Geometrical model of conductor JL/G1A-300/40.**

**Table 2. Structural parameters of JL/G1A-300/40.**

| Layer number | Wire number | Wire diameter (mm) | Cross-section area (mm²) | Lay ratio |
|---|---|---|---|---|
| Core | 1 | 2.66 | 38.9 | - |
| Layer 1 | 6 | 2.66 | | 21 |
| Layer 2 | 9 | 3.99 | 300.9 | 13 |
| Layer 3 | 15 | 3.99 | | 12 |

discretized with 241800 C3D8R elements and the suspension clamp with 14480 C3D8R elements. The mesh convergence is checked and it is demonstrated that the mesh size is fine enough to arrive at accurate convergence. The surface-to-surface contacts are defined to describe the contacts between the wires and the surfaces of the clamp, and all the friction coefficients are set to be 0.1 [21]. The mechanical properties of the steel wire, aluminum wire and the clamp are shown in Table 4. Moreover, the elastoplastic behavior of the steel and aluminum wire is taken into account, and the stress–strain curves of the aluminium and steel wires obtained by experiments [33] are imported into the ABAQUS software.

The left end of the clamp is a symmetric surface, so it is fixed in the FE model. Each end of the conductor segment is coupled with a reference point, the left point is fixed and the right one is specified with loads. The loading processes of the FE model, as illustrated in Fig 19(B), are described as follows.

Step 1: Apply force $F_c$ on the top of the keeper to simulate the pre-tightening force of the bolt, and $F_c$ is calculated by [21]

$$F_c = nT_c/(K_f d_b) \tag{5}$$

where $n = 4$ is the number of bolts; $d_b = 16$ is the nominal bolts diameter, $K_f$ is the thread friction and it is 0.2. Based on the installation standard of the bolts, the torque $T_c$ is set to be 80N·m, and the value of $F_c$ is 10000N determined by Eq (5).

Step 2: Apply axial force $T$ to the right end reference point of the conductor segment to simulate the initial tension in the conductor.

Step 3: Apply the sags, indicated by $S_{sag}$ in Fig 19(B), listed in Table 4 to the right end reference point of the conductor segment.

Step 4: Apply the dynamic displacement $S_{dyn(t)}$ determined by the FE simulation of aeolian vibration of the two-span line in Section 4.1 to the right end reference point of the conductor

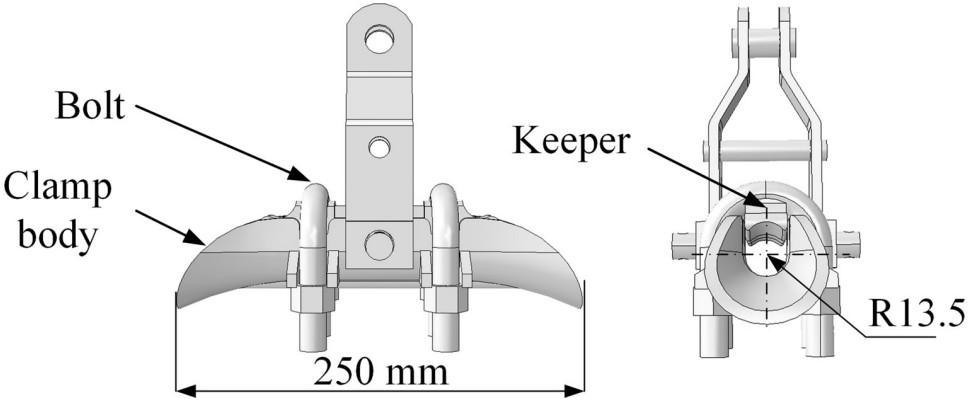

**Fig 17. Dimension of XGU-4F suspension clamp.**

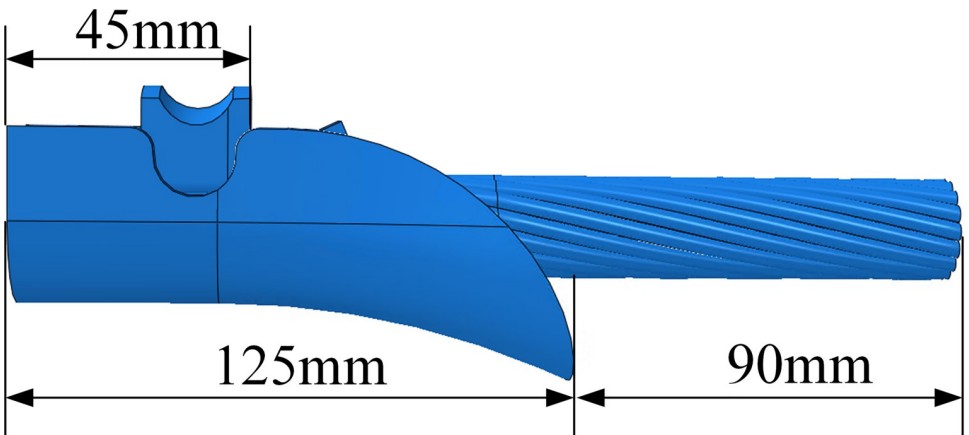

**Fig 18. Analytical model of conductor and suspension clamp.**

segment so as to simulate the aeolian vibration. In steps 1~3, the static analysis is performed and in step 4 the dynamic analysis is carried out in the ABAQUS.

**4.2.3 Stress analysis of conductor segment.** The conductor segment is subjected to alternating bending load during aeolian vibration, and fatigue failure may take place if the aeolian vibration lasts a long time. The maximum stress of the conductor occurs at the time when the applied displacement arrives at the maximum during aeolian vibration in step 4. The stress distribution of the conductor segment and the suspension clamp of case 250 at this time is shown in Fig 20. It is seen that the maximum stress occurs at the left end of the model due to the stress concentration. The stresses on this cross section, including keeper, clamp body, steel wires and aluminum wires, may be larger than the real values because the constrain condition are applied in this cross section.

It can be seen from Fig 20(B) and 20(C) that the stress level of the clamped part of conductor, i.e. the left side part of KE section, is higher, but the stress amplitude during vibration is very small because this part is clamped by the keeper and the clamp body. On the other hand, although the maximum stress of the steel wires is higher than that of the aluminum wires, it is much smaller than its strength. As shown in Fig 13, fatigue failure usually first occurs at the aluminum wires. The maximum stress amplitude of the aluminum wires during aeolian vibration, which determines the fatigue life of the conductor, occurs on the right side part of the keeper edge (KE) section. So, the stress distributions of the aluminum wires of the portion between KE section and Dest section which is shown in Fig 20(C) are shown in Fig 21 and the stress distributions correspond to different cases listed in Table 3 at the time when the maximum stress occurs.

It is seen from Fig 21 that for the lines with the same span length, the larger the initial tension of the conductor, the larger the maximum stress of the aluminum wires during aeolian vibration. For the lines with approximately equal initial tension, the longer the span, the larger the maximum stress of the aluminum wires. It is noted that the maximum stress of the steel

**Table 3. Sags at right end of conductor segment.**

| Case number | 100–1 | 100–2 | 150–1 | 150–2 | 150–3 | 200 | 250 | 300 |
|---|---|---|---|---|---|---|---|---|
| Span length (m) | 100 | 100 | 150 | 150 | 150 | 200 | 200 | 200 |
| Initial tension (MPa) | 84.38 | 90.82 | 61.11 | 80.69 | 90.76 | 86.15 | 81.88 | 72.68 |
| Sag (mm) | 1.56 | 1.48 | 2.91 | 2.41 | 2.22 | 3.07 | 3.98 | 5.18 |

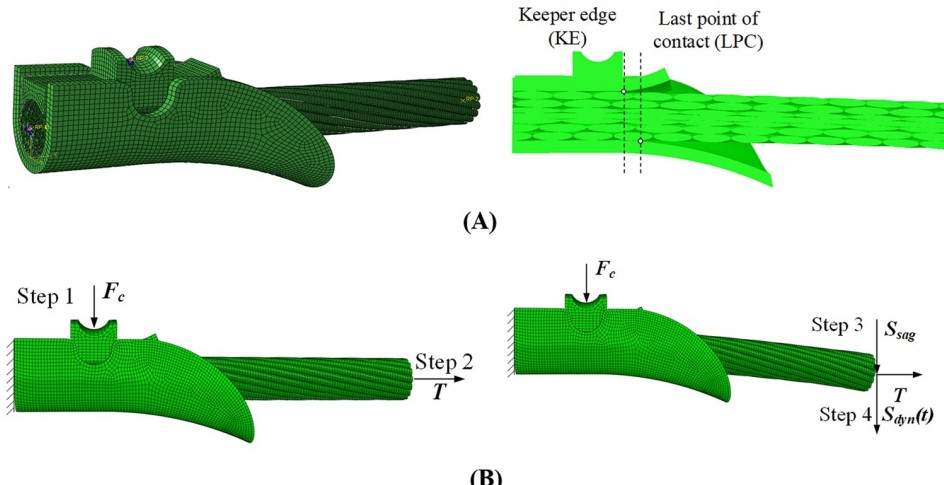

**Fig 19. FE model and loading process of conductor segment and suspension clamp.** (A) Mesh and vertical section of FE model; (B) Loading process.

wires is about 410 MPa and the amplitude of alternating stress is 10 MPa, which results in a much longer fatigue life. And the aluminum wires are more susceptible to fatigue failure than the steel wires [21], so it is unnecessary to consider the fatigue life of the steel wires.

## 4.3. Fatigue life estimation of conductors

Based on the previous work [21], it is known that the aluminum wires of a conductor usually break firstly under aeolian vibration. According to the dynamic responses of the 3D FE model under aeolian vibration, the maximum stresses occur on the outer surface of the inner aluminum wire. The stress amplitudes of some critical points of the conductor segment are selected for fatigue life estimation. The stress histories at 12 typical points of the aluminum wires are extracted, and the peak-to-peak amplitudes and mean values of the stresses are then calculated. The curves of the maximum stress amplitudes versus excitation frequencies in the 8 cases are shown in Fig 22(A).

The influence of the mean stress on the fatigue life of a conductor during aeolian vibration should be taken into account, and it can be introduced by the Goodman equation [23]:

$$\sigma_a/\sigma_{eq} + \sigma_m/\sigma_b = 1 \tag{6}$$

where $\sigma_{eq}$ is the equivalent amplitude of the alternating stress, $\sigma_b$ is the endurance limit, $\sigma_a$ the amplitude of alternating stress and $\sigma_m$ the mean stress. The endurance limit $\sigma_b$ of the aluminum wire of conductor JL/G1A-300/40 is 166 MPa according to our previous work [33]. Substituting the stress amplitudes of the conductors of the 8 cases into Eq (6), the equivalent

**Table 4. Mechanical parameters of suspension clamp and conductor model.**

| Components | Young's modulus (MPa) | Yield stress (MPa) | Ultimate stress (MPa) |
|---|---|---|---|
| Steel wire | 209258 | 1172 | 1489 |
| Aluminum wire | 60928 | 148 | 166 |
| Clamp body and keeper | 152000 | - | - |

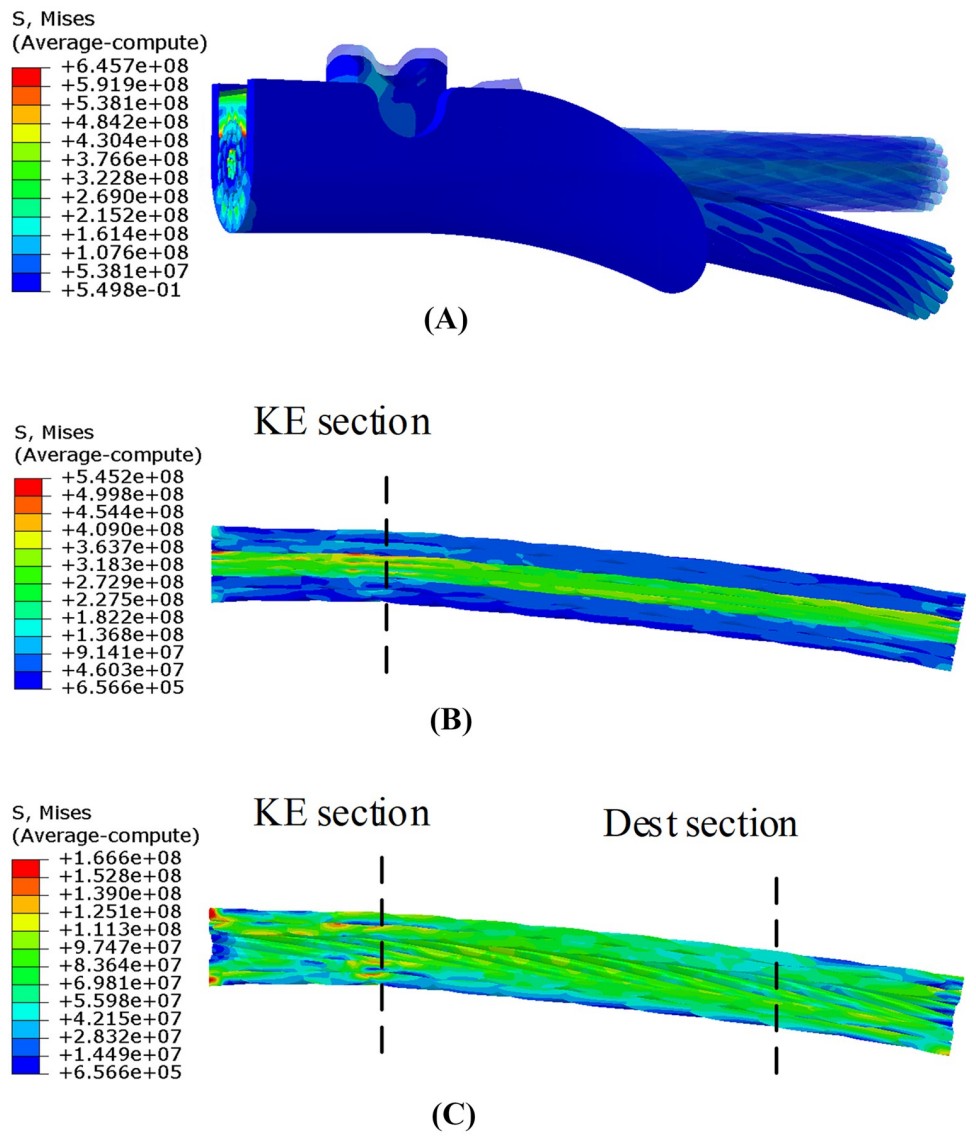

**Fig 20. Stress distributions of 3D model of case 250 (Unit: Pa) (Displacements magnified 8 times).** (A) Conductor segment and the suspension clamp; (B) Steel wires and aluminum wires; (C) Aluminum wires.

stress amplitudes are obtained and shown in Fig 22(B). The stress level is consistent with that in Ref [23].

The fatigue life *N*, the number of vibration cycles, of an aluminum wire can be determined by the *S-N* curve of the safe border line [1]

$$\sigma_{eq} = \begin{array}{l} 450N^{-0.2} \; for \; N \le 1.56 \times 10^7 \\ 263N^{-0.17} \; for \; N > 1.56 \times 10^7 \end{array} \tag{7}$$

If aeolian vibration takes place, the fatigue life of a conductor under wind excitation can be determined by Eq (7), and the results of the 8 cases are shown in Table 5.

It is seen that the fatigue lives of the lines under frequencies in the range of 7.1513 Hz and 8.1029 Hz, the data in column 2 in Table 5, are shortest for all the cases. For example, the

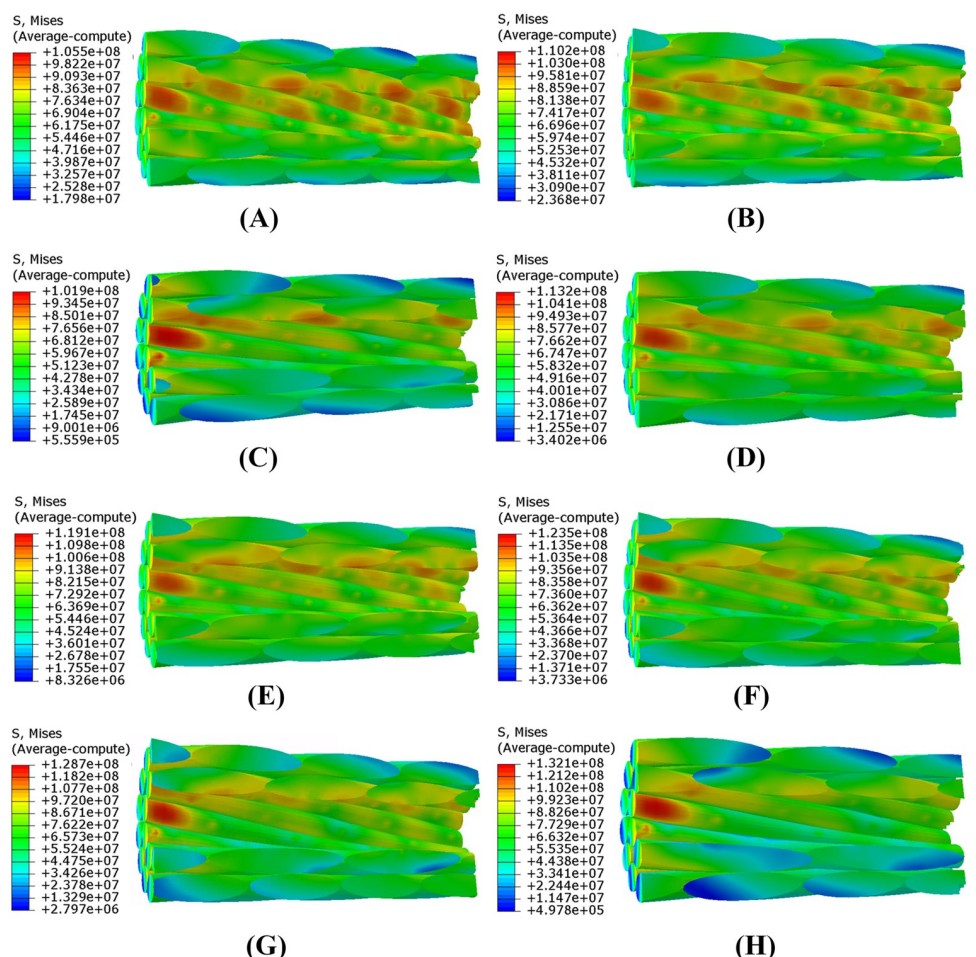

**Fig 21. Stress distributions of aluminum wires of conductor segment (Unit: Pa).** (A) Case 100–1; (B) Case 100–2; (C) Case 150–1; (D) Case 150–2; (E) Case 150–3; (F) Case 200; (G) Case 250; (H) Case 300.

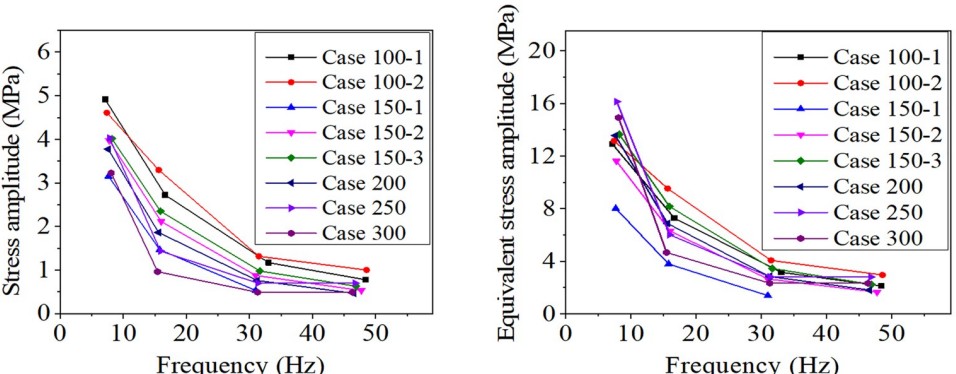

**Fig 22. Maximum stress amplitude and equivalent stress amplitude of conductors versus excitation frequency.** (A) Maximum stress amplitude; (B) Equivalent stress amplitude.

**Table 5. Fatigue lives of conductor lines under different excitation frequencies.**

| Case | Fatigue life $N$ under Excitation frequency $f_e$ | | | |
|---|---|---|---|---|
| **100–1** | $f_e$ = 7.1513 Hz | $f_e$ = 16.6520 Hz | $f_e$ = 33.0347 Hz | $f_e$ = 48.3742 Hz |
| | $4.99{\times}10^7$ | $1.45{\times}10^9$ | $1.94{\times}10^{11}$ | $2.05{\times}10^{12}$ |
| **100–2** | $f_e$ = 7.4161 Hz | $f_e$ = 15.6314 Hz | $f_e$ = 31.4790 Hz | $f_e$ = 48.5590 Hz |
| | $4.48{\times}10^7$ | $2.97{\times}10^8$ | $4.48{\times}10^{10}$ | $2.88{\times}10^{11}$ |
| **150–1** | $f_e$ = 7.6617 Hz | $f_e$ = 15.7822 Hz | $f_e$ = 30.9874 Hz | |
| | $8.24{\times}10^8$ | $6.73{\times}10^{10}$ | $2.47{\times}10^{13}$ | |
| **150–2** | $f_e$ = 7.7628 Hz | $f_e$ = 16.0228 Hz | $f_e$ = 30.9786 Hz | $f_e$ = 47.7241 Hz |
| | $9.31{\times}10^7$ | $3.40{\times}10^9$ | $5.47{\times}10^{11}$ | $8.89{\times}10^{12}$ |
| **150–3** | $f_e$ = 8.2294 Hz | $f_e$ = 15.8880 Hz | $f_e$ = 31.6541 Hz | $f_e$ = 46.8614 Hz |
| | $3.61{\times}10^7$ | $7.39{\times}10^8$ | $1.16{\times}10^{11}$ | $1.49{\times}10^{12}$ |
| **200** | $f_e$ = 7.6120 Hz | $f_e$ = 15.6000 Hz | $f_e$ = 31.2275 Hz | $f_e$ = 46.5629 Hz |
| | $3.75{\times}10^7$ | $2.03{\times}10^9$ | $3.59{\times}10^{11}$ | $5.19{\times}10^{12}$ |
| **250** | $f_e$ = 7.8176 Hz | $f_e$ = 15.9068 Hz | $f_e$ = 31.3561 Hz | $f_e$ = 46.7449 Hz |
| | $1.34{\times}10^7$ | $4.45{\times}10^9$ | $3.95{\times}10^{11}$ | $3.95{\times}10^{11}$ |
| **300** | $f_e$ = 8.1029 Hz | $f_e$ = 15.4526 Hz | $f_e$ = 31.2659 Hz | $f_e$ = 46.3297 Hz |
| | $2.13{\times}10^7$ | $1.98{\times}10^{10}$ | $1.17{\times}10^{12}$ | $1.17{\times}10^{12}$ |

fatigue life of conductor in case 250 with excitation frequency of 7.8176Hz is $1.34{\times}10^7$, which is about 20 days. That means that if aeolian vibration continuously takes place under the same excitation frequency of 7.8176 Hz, the fatigue life of the conductor is only $1.34{\times}10^7$ or 20 days. For the same case, the fatigue life of conductor with excitation frequency of 15.9068 Hz is $4.45{\times}10^9$, which is about 3233 days.

It is noted that conductor life expectancy should be considered to have only qualitative significance [1]. As some lines are installed with dampers, the actual service lives of the above eight conditions still need further investigation.

The aeolian vibration of a conductor line depends on the wind velocity. In real situations, wind velocity changes with the climate of the location of the line passing through, so aeolian vibration may take place intermittently under different excitation frequencies. Therefore, the aeolian vibration of a conductor line depends on the climate change during a year and the estimation of fatigue life is much more complicated. Based on the curves and data in Fig 22 and Table 5, to dwindle aeolian vibration and extend the fatigue life of a conductor line, it may be the most efficient method to control the aeolian vibration with the frequency corresponding to the largest amplitude.

## 5. Discussion

Although wind tunnel tests and the field measurements for aeolian vibration of conductor lines are direct methods, they are too difficult and expensive. The EBP cannot reflect the influences of geometric nonlinearity and span length on the aeolian vibration of a conductor line. The numerical method presented in this study provides an available way to investigate the characteristics of aeolian vibration. From Fig 12, it is seen that the vibration amplitudes determined by the EBP are larger than those by the numerical method. And from Fig 10, the jump phenomenon induced by the nonlinear vibration is reflected by the numerical simulation considering the geometric nonlinearity.

It is very difficult to determine the stress distribution of a conductor during aeolian vibration by means of experiment, especially the stress of the inner layers of the conductor. The FE

method was used to investigate the stress of the conductor line. However, in the existing FE researches, some only used the beam elements to simulate the conductors [19, 20, 22], some did not include the suspension clamp in the model [21], and the others did not take into account the initial bending configuration of the conductor [23]. In this study, a 3D refined model of the conductor segment and the suspension clamp is created, taking into account the initial bending configuration, and the stress distributions of all layers of the conductor during aeolian vibration are obtained.

## 6. Conclusions

In this paper, the CFD model of air flow around a conductor, the FE beam model of conductor line for aeolian vibration and 3D FE model of conductor segment and suspension clamp are set up, and the aeolian vibration characteristics and fatigue lives of conductor lines are investigated. It is concluded that:

1. The CFD can be used to simulate the aerodynamic forces on conductors, and with the FE method of conductor lines discretized with spatial beam elements, the aeolian vibration of the conductor lines can be simulated efficiently.

2. Due to the geometrical nonlinearity of transmission lines, the resonant frequencies for aeolian vibrations are not exactly equal to the linear natural frequencies of the lines, and the jump phenomenon around the resonant frequencies is reflected by the numerical simulation. The steady aeolian vibration is monofrequent, and one lower order of mode is excited at the initial stage of the aeolian vibration and finally disappears.

3. The EBP does not take into account the effect of span length on the aeolian vibration, and it overestimates the amplitudes of aeolian vibration.

4. Based on the 3D FE analysis of the conductor segment and suspension clamp, the maximum stress amplitude of the aluminum wires occurs on the right part of KE, which determines the fatigue life of the conductor. The fatigue failure locations are consistent with those observed in real situations.

5. To dwindle aeolian vibration and extend the fatigue life of a conductor line, it is the most efficient to control the aeolian vibration with the resonant frequency corresponding to the largest vibration amplitude. As some lines are installed with dampers, the actual service lives of aeolian vibration of the transmission conductors need further investigation.

## Supporting information

**S1 Data. The data set includes the original data of figures and can be accessed by the origin software.**
(RAR)

## Author Contributions

**Formal analysis:** Jiaqiong Liu, Zheyue Mou.

**Methodology:** Bo Yan, Getu Niu.

**Software:** Bo Yan, Xiaolin Li.

**Validation:** Yingbo Gao.

**Visualization:** Jiaqiong Liu.

**Writing – original draft:** Jiaqiong Liu, Bo Yan.

**Writing – review & editing:** Jiaqiong Liu, Bo Yan, Zheyue Mou, Yingbo Gao, Getu Niu, Xiao-lin Li.

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
