## [Decision Letter · Decision Letter 0]

23 Nov 2021

PONE-D-21-32631Aeolian vibration characteristics and fatigue life estimation of transmission conductorsPLOS ONE

Dear Dr. Yan,

Thank you for submitting your manuscript to PLOS ONE. After careful consideration, we feel that it has merit but does not fully meet PLOS ONE’s publication criteria as it currently stands. Therefore, we invite you to submit a revised version of the manuscript that addresses the points raised during the review process.

We look forward to receiving your revised manuscript.

Kind regards,

Yasir Nawab, PhD

Academic Editor

PLOS ONE

Journal Requirements:

“The research was funded by the Research Project of Inner Mongolia Power (Group) Co., LTD (No. 2020-39).”

Reviewers' comments:

Reviewer's Responses to Questions

**Comments to the Author**

1. Is the manuscript technically sound, and do the data support the conclusions?

Reviewer #1: Partly

Reviewer #2: Yes

2. Has the statistical analysis been performed appropriately and rigorously? 

Reviewer #1: Yes

Reviewer #2: N/A

3. Have the authors made all data underlying the findings in their manuscript fully available?

Reviewer #1: Yes

Reviewer #2: Yes

4. Is the manuscript presented in an intelligible fashion and written in standard English?

Reviewer #1: Yes

Reviewer #2: Yes

5. Review Comments to the Author

Reviewer #1: Summary

The manuscript titled “Aeolian vibration characteristics and fatigue life estimation of transmission conductors” has been reviewed, keeping in view its suitability, scope of the journal, originality, novelty, comparison with the state of the art, addition to existing literature, methodology and the write-up presentation.

A careful review of the literature reveals that the problem is interesting. The physical model under discussion is not new and the numerical technique applied for the numerical results are academic. Though the scientific values of the results obtained in this paper may be useful (if explained properly) and the simplified model considered here can be taken as a benchmark problem for future experimental research.

Recommendation

The manuscript falls within the scope of the journal and is recommended for publication with minor revision.

General Comments

This reviewer would like the authors to respond on the following fundamental aspects of the work

1. The manuscript should include the discussion on the physical problem under discussion. The readers of the journal are interested to know the addition this particular research brought to the existing literature.

2. Generally, fluids are not stable for higher temperatures, how would this work help to focus on the stability issue of fluids?

3. Thermal resistance should be looked into as well because it’s the main resistance to heat transfer.

4. The abstract, last paragraph of the introduction section and the first paragraph of the conclusion is about the same. Author is recommended to avoid the repetition, it will only contribute to increase the length of the article.

5. Last few lines of abstract to include the main findings of the research carried out in the manuscript.

6. Please provide the assumptions more clearly to facilitate the design engineers to consider when using your model.

7. Introduction is very week, please improve the introduction section using studies only related to the present research. Please update literature by adding the experimental works recently reported on this domain.

8. Author may consider to review the title of the manuscript and made it more specific. Current title is general and unable to explain the specific topic under discussion in the manuscript.

9. There are number of typos (see equation 3) and grammatical error in the presentation of the manuscript and it is recommended authors to read it carefully and correct everything before re-submission.

Reviewer #2: Time varying aerodynamic forces are simulated through 2D CFD models. This research further recognize and argue the impact of various parameters defining the characterization behavior of the system. In my opinion, the article is suitable for publication. However, below are few minor observations and suggestions which I hope will be helpful in improving the quality of the manuscript.

1 - The results are narrated in very simple manner by just reading the numerical outcomes. I think physical justification of the outcomes will be value addition and will enhance the elegance of the research.

2 - The units reported are not consistent. For example, in line 157 size of domain is reported as 10m x 10 m (first one is without space and second include space), both should be in same style, I believe standard formation is to report unit with space. This is observed throughout the manuscript.

3 - Section 2.1.1 does not include the justification of the choice of parameters governing simulations lines 155-157.

4- Equation 3 is not fully explained and there are few typos as well.

Hopefully the afore-mentioned suggestions will be helpful.

6. PLOS authors have the option to publish the peer review history of their article (what does this mean?). If published, this will include your full peer review and any attached files.

Reviewer #1: No

Reviewer #2: No

---

## [Author Response · Author response to Decision Letter 0]

8 Dec 2021

Response to reviewers:

The authors thank the academic editor and the reviewers very much for their good suggestions, and have tried to respond to the comments and revise the manuscript carefully. The responses to the comments are listed as the follows one by one.

Responses to comments of reviewer 1

The manuscript titled “Aeolian vibration characteristics and fatigue life estimation of transmission conductors” has been reviewed, keeping in view its suitability, scope of the journal, originality, novelty, comparison with the state of the art, addition to existing literature, methodology and the write-up presentation.

A careful review of the literature reveals that the problem is interesting. The physical model under discussion is not new and the numerical technique applied for the numerical results are academic. Though the scientific values of the results obtained in this paper may be useful (if explained properly) and the simplified model considered here can be taken as a benchmark problem for future experimental research.

Recommendation

The manuscript falls within the scope of the journal and is recommended for publication with minor revision.

General Comments

This reviewer would like the authors to respond on the following fundamental aspects of the work.

1. The manuscript should include the discussion on the physical problem under discussion. The readers of the journal are interested to know the addition this particular research brought to the existing literature.

Response: Thank the reviewer very much for his/her good suggestions. A new section is inserted to discuss the physical problem and to let the readers know the addition this research brought to the existing work. Please see Section 5. Discussion. 

2. Generally, fluids are not stable for higher temperatures, how would this work help to focus on the stability issue of fluids?

Response: In general, the temperatures in transmission conductors in service are less than 70℃, so the air flow with different temperatures including 20, 30, 40, 50, 60 and 70℃ around the conductor are analyzed. In all the cases the stable vortex streets and alternating aerodynamic forces are observed. The amplitudes of the lift coefficient corresponding to the six temperatures are 1.35, 1.32, 1.28, 1.25, 1.22 and 1.19 respectively. It is seen that the amplitude of the lift coefficient at 70℃ is about 11% smaller than that at 20℃. It is very difficult to determine the temperature of the air around the conductor in real situation, but the temperature may be much lower than 70o. The aerodynamic forces in the case of 20℃ are used in this paper. And, the fluids are assumed to be stable. A marked sentence is inserted into the first paragraph of Section 2.1.1.

3. Thermal resistance should be looked into as well because it’s the main resistance to heat transfer.

Response: We agree with the reviewer’s suggestion. Thermal resistance should be considered when considering the heat transfer between the transmission lines and the surroundings. But, based on work (Zhang et al. 2021), the fatigue life of an energized conductor with higher temperature is slightly larger than that of a nonelectrical conductor with lower temperature. In addition, the mechanical characteristics of the transmission conductor under aeolian vibration are mainly focused on in this study. Thus, the influences of heat transfer of transmission conductor and the surroundings on the vibration characteristics and fatigue lives of the conductors are ignored. And the thermal resistance is not considered in this paper. Anyway, this is a good suggestion and the authors will consider this in the future. Few sentences are inserted into the first paragraph in Section 4.2.1 to explain our ideas in detail. 

Reference: Zhang M, Zhou J, Zhao GF, Xu JK, Sun C. Study of structural-thermal characteristics of electrified conductors under aeolian vibration. Wind Struct. 2021; 33(2): 155-168. http://doi.org/10.12989/was.2021.33.2.155

4. The abstract, last paragraph of the introduction section and the first paragraph of the conclusion is about the same. Author is recommended to avoid the repetition, it will only contribute to increase the length of the article.

Response: Thank the reviewer very much for the careful and constructive suggestions. The last paragraph of the introduction section and the first paragraph of the conclusion are revised. Please see the relevant contents in the manuscript. 

5. Last few lines of abstract to include the main findings of the research carried out in the manuscript.

Response: Thank the reviewer very much. The main findings of the research are added in the abstract. Please see the marked sentences in the abstract.

6. Please provide the assumptions more clearly to facilitate the design engineers to consider when using your model.

Response: Thank the reviewer very much for the thoughtful consideration of the manuscript. The assumptions are added into the manuscript which can be seen in the first paragraph in Section 2.2.

7. Introduction is very weak, please improve the introduction section using studies only related to the present research. Please update literature by adding the experimental works recently reported on this domain.

Response: Thank the reviewer very much. Some literatures (Xia et al. 2013; Qi et al. 2019; Heics et al. 1994) that are not fully related to this research are deleted and some recently experimental works (Goudreau et al. 2010; Kubelwa et al. 2020; Kubelwa et al. 2016; Reinke et al. 2019; Miranda et al. 2021) that are related to this research are added. Some explanations are inserted into the introduction. Please see the marked sentences in the introduction. 

References:

The deleted literatures:

[1] Xia YP, Rui XM, Yang L, Wang DD, Liang JY. CFD numerical simulation of aeolian vibration of single conductor based on the ADINA. Appl. Mech. Mater. 2013; 455: 292–297. http://doi.org/10.4028/www.scientific.net/AMM.455.292

[2] Qi Y, Rui X, Ji K, Liu C, Zhou C. Study on aeolian vibration suppression schemes for large crossing span of ultra-high-voltage eight-bundle conductors. Adv. Mech. Eng. 2019; 11(4): 1-12. http://doi.org/10.1177/1687814019842706

[3] Heics RC, Havard DG. Influence of vibration recorders on conductor vibration. IEEE Trans. Power Deliv. 1994; 9: 919–938. http://doi.org/10.1109/61.296275

The added literatures (which are listed with No. 9, 11, 12, 13 and 14 in the References of the manuscript):

[4] Goudreau S, Levesque F, Cardou A, Cloutier L. Strain measurements on ACSR conductors during fatigue tests II—stress fatigue indicators. IEEE Trans. Power Deliv. 2010; 25: 2997–3006. http://doi.org/10.1109/TPWRD.2010.2042083

[5] Kubelwa YD, Swanson AG, Dorrell DG. Aeolian vibrations of overhead transmission line bundled conductors during indoor testing, Part B: assessment of fatigue and damping performances. CIGRE. Cigre Science & Engineering. N°18 June 2020.

[6] Kubelwa YD, Loubser RC, Papailiou KO. Statistical modelling of bending stress in ACSR overhead transmission line conductors subjected to aeolian vibrations-I. In Proceedings of the World Congress on Engineering, London, U.K., 29 June–1 July. 2016.

[7] Reinke G, Badibanga RK, Pestana MS, Ferreira JL, Araujo JA, Araujo CRM. Failure analysis of aluminum wires in all aluminum alloy conductors–AAAC. Eng. Fail. Anal. 2020; 107: 104197. http://doi.org/10.1016/j.engfailanal.2019.104197

[8] Miranda T, Badibanga R, Araújo JA, Silva C, Ferreira J. Fatigue evaluation of all aluminium alloy conductors fitted with elastomeric and metallic suspension clamps. IEEE Trans. Power Deliv. 2021; PP(99): 1-1. http://doi.org/10.1109/TPWRD.2021.3064823

8. Author may consider to review the title of the manuscript and made it more specific. Current title is general and unable to explain the specific topic under discussion in the manuscript.

Response: The title of the manuscript is revised as “Numerical study of aeolian vibration characteristics and fatigue life estimation of transmission conductors”.

9. There are number of typos (see equation 3) and grammatical error in the presentation of the manuscript and it is recommended authors to read it carefully and correct everything before re-submission.

Response: Thank the reviewer very much. The authors read the manuscript carefully and corrected some errors, including the typo of equation 3.

Responses to comments of reviewer 2

Time varying aerodynamic forces are simulated through 2D CFD models. This research further recognize and argue the impact of various parameters defining the characterization behavior of the system. In my opinion, the article is suitable for publication. However, below are few minor observations and suggestions which I hope will be helpful in improving the quality of the manuscript.

1 - The results are narrated in very simple manner by just reading the numerical outcomes. I think physical justification of the outcomes will be value addition and will enhance the elegance of the research.

Response: Thank the reviewer for these precious comments and suggestions. Yes, the physical justification of the numerical outcomes will be more valuable and enhance the elegance of the research. However, the CFD model of air flow around a conductor and the FE model of a conductor under aeolian vibration have been verified by the previous works. It is very difficult to determine the stress distribution of a conductor during aeolian vibration by means of experiment, especially the stress of the inner layers of the conductor. And the aeolian vibration test in real situation is too difficult, which may not be completed in a short time. Thus, numbers of numerical simulations were carried out to analyze the aeolian vibration characteristics and fatigue lives of the transmission conductors in detail. We think, the detailed experiment work will be focused on by more efforts in the next work. Two sentences are added into the beginning of the first paragraph and the second paragraph of Section 5 respectively.

In addition, some physically explanations and analyses for the results show in the following. A marked sentence is inserted into lines 6-8 of the second paragraph of Section 3.2, which compares the aeolian vibration characteristic with the Ref. [5]. A marked sentence is inserted into lines 5-7 of the first paragraph of Section 3.3, which compares and explains the variation of the vibration amplitude with frequency in this study. A sentence is added into the end of the second paragraph of Section 4.3, which compares the stress level with the Ref. [24]. And conclusion (4) is verified with Fig. 13 which is a fatigue failure of aluminum wires in conductor in real situations.

2 - The units reported are not consistent. For example, in line 157 size of domain is reported as 10m x 10 m (first one is without space and second include space), both should be in same style, I believe standard formation is to report unit with space. This is observed throughout the manuscript.

Response: Thank the reviewer very much for the careful review of the manuscript. Yes, the standard formation is to report unit with space. The manuscript are advised with this standard.

3 - Section 2.1.1 does not include the justification of the choice of parameters governing simulations lines 155-157.

Response: The justification of the choice of the parameters governing simulations lines 155-157 is explained in the manuscript. Sentence 1 “Aeolian vibration is generated by moderate wind in the range 0.5 to 7 m/s and it is most likely to occur when the wind direction is perpendicular to the conductor lines [1].” and sentence 2 “Too large size of the domain will increase the computation cost and too small size will affect the air flow around the conductor. After several trials, it is found the size of the domain set to be 10 m × 10 m is suitable.” are added into the second paragraph of Section 2.1.1.

4- Equation 3 is not fully explained and there are few typos as well.

Response: Equation 3 is fully explained and the typos of which are corrected. It can be seen in the penultimate paragraph in Section 2.2. Thank you!

---

## [Decision Letter · Decision Letter 1]

13 Jan 2022

Numerical study of aeolian vibration characteristics and fatigue life estimation of transmission conductors

PONE-D-21-32631R1

Dear Dr. Yan,

We’re pleased to inform you that your manuscript has been judged scientifically suitable for publication and will be formally accepted for publication once it meets all outstanding technical requirements.

Kind regards,

Yasir Nawab, PhD

Academic Editor

PLOS ONE

Additional Editor Comments (optional):

Reviewers' comments:

Reviewer's Responses to Questions

**Comments to the Author**

1. If the authors have adequately addressed your comments raised in a previous round of review and you feel that this manuscript is now acceptable for publication, you may indicate that here to bypass the “Comments to the Author” section, enter your conflict of interest statement in the “Confidential to Editor” section, and submit your "Accept" recommendation.

Reviewer #1: All comments have been addressed

Reviewer #2: All comments have been addressed

2. Is the manuscript technically sound, and do the data support the conclusions?

Reviewer #1: Yes

Reviewer #2: Yes

3. Has the statistical analysis been performed appropriately and rigorously? 

Reviewer #1: Yes

Reviewer #2: N/A

4. Have the authors made all data underlying the findings in their manuscript fully available?

Reviewer #1: Yes

Reviewer #2: (No Response)

5. Is the manuscript presented in an intelligible fashion and written in standard English?

Reviewer #1: Yes

Reviewer #2: Yes

6. Review Comments to the Author

Reviewer #1: All the comments of the reviewer are addressed in the revised manuscript. It is now recommended for publication in the journal PLOS One

Reviewer #2: The revised manuscript addresses all the quires of the reviewer thefore i recommend this article for publication.

7. PLOS authors have the option to publish the peer review history of their article (what does this mean?). If published, this will include your full peer review and any attached files.

Reviewer #1: No

Reviewer #2: No

---

## [Editor Report · Acceptance letter]

17 Jan 2022

PONE-D-21-32631R1 

Numerical study of aeolian vibration characteristics and fatigue life estimation of transmission conductors 

Dear Dr. Yan:

I'm pleased to inform you that your manuscript has been deemed suitable for publication in PLOS ONE. Congratulations! Your manuscript is now with our production department. 

Kind regards, 

on behalf of

Dr. Yasir Nawab 

Academic Editor

PLOS ONE